# RUNX3 in Stem Cell and Cancer Biology

**DOI:** 10.3390/cells12030408

**Published:** 2023-01-25

**Authors:** Linda Shyue Huey Chuang, Junichi Matsuo, Daisuke Douchi, Nur Astiana Bte Mawan, Yoshiaki Ito

**Affiliations:** 1Cancer Science Institute of Singapore, NUS Centre for Cancer Research, Yong Loo Lin School of Medicine, National University of Singapore, 14 Medical Drive #12-01, Singapore 117599, Singapore; 2Department of Surgery, Tohoku University Graduate School of Medicine, Sendai 980-8574, Japan

**Keywords:** RUNX3, stem cells, cancer, cell cycle, proliferation, differentiation block, early-stage cancer

## Abstract

The runt-related transcription factors (RUNX) play prominent roles in cell cycle progression, differentiation, apoptosis, immunity and epithelial–mesenchymal transition. There are three members in the mammalian RUNX family, each with distinct tissue expression profiles. RUNX genes play unique and redundant roles during development and adult tissue homeostasis. The ability of RUNX proteins to influence signaling pathways, such as Wnt, TGFβ and Hippo-YAP, suggests that they integrate signals from the environment to dictate cell fate decisions. All RUNX genes hold master regulator roles, albeit in different tissues, and all have been implicated in cancer. Paradoxically, RUNX genes exert tumor suppressive and oncogenic functions, depending on tumor type and stage. Unlike RUNX1 and 2, the role of RUNX3 in stem cells is poorly understood. A recent study using cancer-derived RUNX3 mutation R122C revealed a gatekeeper role for RUNX3 in gastric epithelial stem cell homeostasis. The corpora of *RUNX3^R122C/R122C^* mice showed a dramatic increase in proliferating stem cells as well as inhibition of differentiation. Tellingly, *RUNX3^R122C/R122C^* mice also exhibited a precancerous phenotype. This review focuses on the impact of RUNX3 dysregulation on (1) stem cell fate and (2) the molecular mechanisms underpinning early carcinogenesis.

## 1. Introduction

The mammalian RUNX transcription factor family comprises three major developmental regulators, namely RUNX1, RUNX2 and RUNX3. All RUNX proteins share the evolutionarily conserved DNA-binding Runt domain at the N-terminus. The Runt domain heterodimerizes with CBFβ to bind stably to the DNA motif 5′-ACCRCA-3′. Intriguingly, the three mammalian RUNX paralogs show different binding affinities for the consensus motif [1]. While the C-terminal domain is less well conserved among the RUNX members, it invariably contains the transactivation domain as well as protein interaction domains, such as the PY and VWRPY motifs, which bind transcriptional coactivator YAP and co-repressor Groucho/TLE, respectively [2]. Therefore, depending on interacting proteins, RUNX may activate or repress genes. Moreover, because they bind the same DNA sequence, RUNX proteins may serve redundant or antagonistic roles, as well as unique roles.

Early studies using *Runx* knockout (KO) mice have indicated that *RUNX* genes are critically involved in developmental processes of diverse tissue types. *RUNX1* is required for developmental hematopoiesis—homozygous *Runx1* KO mice were unable to generate hematopoietic stem cells and showed embryonic lethality [3]. *RUNX2* is a master regulator of bone development and is necessary for mesenchymal stem cells to differentiate into osteoblasts—homozygous *Runx2* KO mice exhibited severe bone malformation, dying shortly after birth because of breathing disability [4,5]. Homozygous *Runx3* KO mice died soon after birth, likely by starvation [6]. Their stomach mucosae were considerably thicker than wildtype counterparts, and this has been attributed to increased gastric epithelial cell proliferation and suppressed apoptosis [6]. Moreover, *RUNX3* is involved in sensory neuron differentiation and has been shown to regulate the axonal projection of proprioceptive dorsal root ganglion (DRG) neurons—*Runx3* KO mice showed severe limb ataxia and abnormal posture [7].

Both tumor suppressive and oncogenic roles have been ascribed to *RUNX* genes [8]. Recurrent mutations in the Runt domain of *RUNX1* have been identified in acute myeloid leukemia and luminal-type breast cancer [9,10,11]. While loss-of-function mutations suggest a tumor suppressor role for *RUNX1*, *RUNX1* was reported to reinforce the TAL1-driven oncogenic program in T-cell acute lymphoblastic leukemia [12]. Moreover, conditional *Runx1* knockout mice have indicated the importance of *RUNX1* for tumor formation in hair follicle stem cells [13]. The high expression levels of RUNX1 in skin squamous cell carcinoma, esophageal, lung, colon and pancreatic cancers indicate that *RUNX1* may drive oncogenesis in various solid tumors [13]. *RUNX2* is overexpressed in osteosarcoma, breast and prostate tumors, as well as cells that metastasize to the bone [14]. Conditional *Runx2* KO mice revealed roles for *Runx2* in regulation of epithelial cell fate in mammary gland development and breast cancer [15]. *Runx3*-deficient mice are predisposed to cancers of the breast, lung, and gastrointestinal tract [16]. *RUNX3* is frequently inactivated by hypermethylation and protein mislocalization in solid tumors [17,18,19]. In this review, we summarize the impact of RUNX3 dysregulation on stem cell self-renewal/differentiation and epithelial cancer development. We discuss how RUNX3 serves as gatekeeper via its interactions with oncogenic signaling pathways and identify questions that may lead to new insights on cancer stem cell biology.

## 2. Stem Cell Regulation Is a Core Conserved RUNX Function

It is interesting that the sole *Runx* gene in the nematode *Caenorhabditis elegans rnt-1* plays critical roles in regulating proliferation, self-renewal and differentiation of the stem-like seam cells [20,21,22]. Overexpression of *rnt-1* and CBFβ homologue *bro-1* resulted in seam cell hyperplasia and concomitant reduction of differentiated cells [20,23]. Conversely, rnt-1 or bro-1 deficiencies resulted in defective seam cell divisions and, thus, reduction of seam cell populations [20,23]. *rnt-1* also cooperates with the Wnt signaling pathway to regulate asymmetrical cell division of T blast cells [22]. rnt-1 mutants show loss of polarity in the asymmetrical T cell division.

*C. elegans* diverged from vertebrates early in metazoan development. The fact that *RUNX* genes in both *C. elegans* and humans regulate stem cells indicates that this particular RUNX property arose early during metazoan evolution and that stem cell regulation might be a principal function of ancestral RUNX. In mammals, adult stem cells have been heavily implicated in cancer initiation and progression. From studies on *RUNX1* and *RUNX2*, it would seem that deregulated *RUNX* genes are causally involved in stem cell dysfunction, be it through hyperproliferative stem cells, aberrant cell division and/or differentiation blocks [13,15,24,25,26,27,28]. Here, we review how *Runx3*-deficent mouse models reveal the roles of *RUNX3* in epithelial homeostasis, in particular stem cell renewal and differentiation.

## 3. Regulation of *RUNX3* Gene

The *RUNX3* gene is regulated by two promoters, namely the distal P1 and proximal P2 [29,30]. Transcripts from the P1 and P2 promoters give rise to RUNX3 isoforms that differ only at the extreme N-terminal region [29,30]. The P1 promoter contains two RUNX consensus motifs [29,30], which suggest auto-regulation, as well as cross-regulation, by RUNX paralogs. The P1 promoter is responsible for the high *RUNX3* expression levels in CD8+ T and T_H_1 cells [31]. The P2 promoter contains a large CpG island [29], which is frequently hypermethylated and epigenetically silenced in solid tumors [17,32]. Transcripts expressed from the P2 promoter are inefficiently expressed, relative to P1 [33]. While high *RUNX3* expression in lymphocytes indicates its important roles in T-cell maturation [34,35], the emerging view is that the relatively lower *RUNX3* expression in epithelial cells is necessary for tissue homeostasis and that perturbations in *RUNX3* expression might alter the balance between proliferation and differentiation. Indeed, aberrant *RUNX3* expression is associated with epithelial tumorigenesis [17,36] (Figure 1).

The regulation of the *RUNX3* gene is not well studied. Polycomb repressive complex 2 components, such as EZH2 and SUZ12, are central to stem cell function during early development [37,38]. *RUNX3* is one of the developmental regulator genes bound by SUZ12 in human embryonic stem cells [39]. EZH2 has been shown to repress RUNX3 transcription in cancer cell lines [40]. While *EZH2* is highly expressed in embryonic stem cells and plays important roles in stem cell maintenance, elevated *EZH2* expression has also been observed in multiple cancers, in particular, the more primitive and malignant types [37]. While highly speculative, it maybe that oncogenic EZH2 induces epigenetic silencing of *RUNX3* expression to promote dedifferentiation and, thus, cellular plasticity.

The frequent epigenetic silencing of RUNX3 in multiple solid tumor types indicates a strong RUNX3 gatekeeper role during early-stage cancer, while the abnormally elevated *RUNX3* expression in pancreatic ductal adenocarcinoma indicates a role—in conjunction with TGFβ pathway component *SMAD4/DPC4*—in directing a metastatic transcriptional program [17,36]. Although rare, RUNX3 missense mutations have been identified in cancer patients. In particular, the R122C mutation in Runt domain has been instrumental in understanding RUNX3′s tumor suppressor roles [6,41,42].

## 4. Insights from *Runx3* KO Mice: Interactions with Oncogenic Signaling Pathways

Mouse knockout models have been instrumental in understanding the role of RUNX3 during carcinogenesis. While it should be noted that *Runx3* deficiency in different mouse strains (e.g., BALB/c and C57BL/6) demonstrated phenotypic variability with respect to the inflammatory response, all revealed susceptibility to preneoplastic changes. RUNX3 modulates the signaling intensities of developmental pathways, such as the transforming growth factor β (TGFβ), wingless-type MMTV integration site (Wnt) and RAS signaling pathways [8]. The TGFβ pathway regulates proliferation, differentiation and apoptosis and, as such, plays crucial roles in tissue homeostasis and regeneration [43]. TGF-β signaling contributes to the maintenance and differentiation of various tissue stem cell types through intrinsic signaling, as well as non-autonomous cues from niche cells [44]. TGFβ signaling is context-dependent during tumorigenesis, functioning as a tumor suppressor in preneoplasia and metastasis promoter in late-stage cancer [43]. Early studies revealed the interaction and functional cooperation of RUNX3 with TGFβ effectors SMAD2/3 [45,46] (Figure 1). RUNX3 binds to the RUNX consensus motifs in the promoters of cell cycle inhibitor *CDKN1A* and pro-apoptotic *BIM* to directly regulate their transcription in a TGFβ-dependent manner [47,48]. The gastric mucosa of the *Runx3* null C57BL/6 mouse model has been shown to exhibit hyperplasia, which was attributed to increased proliferation and suppressed apoptosis as a result of impaired TGFβ signaling [6]. Gastric epithelial cell lines derived from the fetal stomach of *Runx3^−/−^ p53^−/−^* mouse are prone to spontaneous epithelial–mesenchymal transition (EMT) and this results in a tumorigenic stem-cell-like subpopulation [49,50]. It has also been suggested that RUNX3 safeguards gastric epithelial cells from aberrant Wnt- and TGFβ-mediated cellular plasticity and stemness [49].

RUNX3 has been reported to attenuate oncogenic Wnt signaling, independent of *adenomatous polyposis coli (Apc)*, in the gastrointestinal tract [51] (Figure 1). *Runx3*-deficient mice exhibited elevated Wnt activity, accompanied by increased proliferation in the intestine [51]. Biallelic inactivation of *RUNX3* induced colon adenomas, which indicates a gatekeeper role for *RUNX3* in colon adenoma development [51]. Mechanistically, RUNX3 forms a ternary complex with Wnt effectors β-catenin/TCF4 through their respective DNA binding domains, namely Runt and HMG (high-mobility group) [51]. This interaction impairs the DNA binding ability of the β-catenin/TCF4 complex, thereby suppressing intestinal oncogenic Wnt signaling [51]. *APC* is a key negative regulator of the Wnt pathway. It provides a scaffold for the β-catenin destruction complex and is important for rapid β-catenin turnover [52]. The high frequency of *APC* gene mutations in colorectal cancers suggests that *APC* dysfunction and subsequent elevated Wnt activity are early and/or initiating events in colorectal cancer [52]. Thus, *Runx3*-deficient mice were compared with the *Apc^Min/+^* mouse model, which harbors a dominant loss of function mutation at the *Apc* gene. At 65 weeks of age, the frequency of adenoma development in the small intestine of *Runx3^+/−^* mice was comparable to that of *Apc^Min/+^* mice with the same BALB/c background [51].

Similarly, the loss of *Runx3* is associated with premalignant changes in the gastric corpus epithelium of BALB/c mice [42]. The gastric epithelia in *Runx3^−/−^* adult mice were hyperplastic with higher proliferation rates [42]. The carcinogen N-methyl-N-nitrosourea readily induced adenocarcinomas in *Runx3^−/−^* mice, unlike the wildtype mice [42]. While the acid-secreting parietal cell population was not affected, there was a distinct loss of digestive enzyme-secreting, terminally differentiated chief cells and the development of an intestinal phenotype, as marked by the expression of intestinal transcription factor *Cdx2* [42]. *Runx3* is, therefore, potentially important for chief cell differentiation and its loss may promote chief cell dedifferentiation [42]. Moreover, *Runx3* deficiency resulted in upregulation of Wnt targets, such as *Axin2*, *Myc* and *CD44*, suggesting enhanced Wnt activity. It was reported that after stomach corpus tissue injury and subsequent Wnt activation, the subpopulation of chief cells expressing Wnt target *Lgr5* functioned as ‘reserve’ stem cells to promote epithelial renewal [53]. Moreover, *Lgr5*-expressing chief cells were identified to be a major cell of origin of gastric cancer [53]. It is reasonable to propose that *Runx3* deficiency led to Wnt-related chief cell plasticity and a precancerous phenotype. Chronic *H. pylori* infection, a class one carcinogen in gastric cancer, has been reported to promote *RUNX3* promoter hypermethylation [54] and its subsequent inactivation in gastric cancer. Notably, RUNX3 directly activates the transcription of one of the key regulators of innate immunity, *IL23A* [55]. This ability to upregulate *IL23A* expression is strongly enhanced by TNF-α/NF-kB stimulation and *H. pylori* infection, thus indicating the involvement of RUNX3 during infection of gastric epithelial cells and its potential protective roles in the inflammatory response and pathogen clearance [55]. *RUNX3* inactivation might be one of the main mechanisms through which *H. pylori* promotes carcinogenesis [56]. The observation that *RUNX3* hypermethylation occurs more frequently in intestinal-type, relative to diffuse-type, gastric carcinomas, suggests that *RUNX3* functions as a gatekeeper of intestinal-type gastric carcinomas [32].

Aside from the gastrointestinal tract, *RUNX3* is frequently silenced by epigenetic methylation in breast cancer [57]. In total, 20% of *Runx3^+/−^* BALB/c female mice developed ductal carcinoma [58]. Mechanistically, RUNX3 inhibits estrogen-dependent proliferation and oncogenic potential of ERα-positive breast cancer cells by reducing the stability of ERα protein [58]. Increased ERα stability is one of the main reasons why ERα is upregulated in 70% of breast cancer [59]. The ability of RUNX3 to modulate ERα activity indicates a strong tumor suppressor role for RUNX3 in breast cancer (Figure 1).

*Runx3* KO mice also revealed a strong RUNX3 gatekeeper role in *Ras*-induced lung tumorigenesis [60]. *Runx3* inhibits adenoma formation in the lung and its inactivation is an early event in lung adenocarcinoma formation [60]. The specific interaction of RUNX3 with bromodomain-containing protein 2 (BRD2) in the early phase of the cell cycle indicates its involvement in regulating cell cycle entry [60]. In the presence of oncogenic *Ras*, RUNX3 cooperates with BRD2 to activate the ARF-p53 pathway and promote apoptosis [60]. Bromodomain proteins are considered to be chromatin ‘readers’, which recruit enzymes that modify chromatin to regulate gene expression [61]. How RUNX3 influences chromatin modelling during the various phases of the cell cycle remains to be determined.

## 5. Interactions of RUNX3 with the Cell Cycle

In the cell cycle, the restriction (R) point is when cells decide whether to proliferate or remain in quiescence. Proper regulation of the R-point is essential for S phase entry and normal differentiation. RUNX3 regulates the R point through its interactions with BRD2 and the tumor suppressor retinoblastoma protein pRB, and subsequent induction of *CDKN1A* expression [60,62,63] (Figure 1). The *retinoblastoma susceptibility gene (RB)* is frequently mutated in a wide range of cancer types [64]. While pRB is best known—through its inhibition of E2F transcription factors—for its role in regulating G1-S transition, it is now considered to be a transcription co-factor that binds and modulates the activities of numerous transcription factors, as well as an adaptor protein that recruits chromatin modelling enzymes to target genes [64]. pRB has been reported to regulate differentiation and maintain permanent cell cycle arrest, as well as chromosomal stability [64]. *CDKN1A* was shown to regulate stem cell kinetics through its control of stem cell entry into the cell cycle—its maintenance of quiescence in hematopoietic stem cells prevents premature stem cell depletion [37,65]. Aside from the R-point, the fact that RUNX3 remains associated with chromosomes during mitosis suggests potential for epigenetic control of cellular memory, perhaps during lineage specification [66]. RUNX3 has also been detected at the mitotic machinery, such as midbody, mitotic spindle and centrosome [67,68]. Its interactions with Aurora kinases and the fact that knockdown of RUNX3 delayed mitotic entry indicates a regulatory role at the G2/M checkpoint [67,68]. Asymmetric division balances the stem cell population and differentiating progeny cells to maintain proper tissue homeostasis [69]. The older centrosome is inherited by the stem cell during asymmetric cell division [70]. Post-abscission midbodies, which associate with the older centrosome, are preferentially enriched in stem cells and cancer cells, where they may enhance reprogramming and increase tumorigenicity, respectively [70]. It remains to be seen whether RUNX3, at its vantage points in the centrosome and midbody, contributes to this important aspect of stem cells. Moreover, although speculative, the RUNX3-Wnt association [51] suggests that mammalian stem cells may recapitulate the *rnt1*-Wnt cooperation seen in the asymmetric division of seam cells in *C. elegans* [22].

## 6. Clues from the *RUNX3^R122C^* Knock-In Mouse Model

We previously identified the RUNX3 single missense mutation R122C from a human gastric cancer patient [6]. Since then, the *RUNX3^R122C^* mutation has been described in head and neck and cervical squamous carcinomas (cbioportal database; http://www.cbioportal.org; accessed on 22 August 2022). Runt domain mutant RUNX3^R122C^ is impaired in binding to the RUNX consensus sequence and, thus, is unable to transcriptionally activate the cell cycle inhibitor *CDKN1A* [48]. Not surprisingly, while wildtype RUNX3 exerts strong growth inhibition, the RUNX3^R122C^ mutant is associated with increased cell proliferation [6]. In addition, the RUNX3^R122C^ mutant protein showed drastically reduced interactions with oncogenic transcription factors TCF4 and TEAD4, failing to suppress their oncogenic activities [51,71]. This impaired ability to inhibit TCF4 and TEAD4 indicates that the RUNX3^R122C^ mutation—independent of its altered affinity for DNA—might have significant consequences on Wnt and TEAD signaling activities. It is currently unknown whether the RUNX3^R122C^ mutant binds sequences other than the consensus RUNX sequence. Moreover, it remains to be seen whether the R122C mutation affects RUNX3 protein stability and binding to CBFβ.

To understand the effects of the *RUNX3^R122C^* mutation on stem cell homeostasis and gastric carcinogenesis, we generated the *RUNX3^R122C^* knock-in C57BL/6 mouse model and studied the corpus gland of the stomach [41]. The corpus gland comprises four regions—the pit, isthmus, neck and base. Proliferating stem cells reside in the isthmus region while the base region consists of non-proliferative differentiated chief cells, as well as ‘reserve’ stem cells. *Runx3* mRNA expression was detected in the epithelial cells of both the isthmus and base regions [41]. This observation suggests functional roles for *Runx3* at both regions. At 6 months of age, the *RUNX3^R122C/R122C^* homozygous mice exhibited a precancerous phenotype known as spasmolytic-polypeptide-expressing metaplasia (SPEM) in the stomach corpus [41]. The elongated fundic metaplastic mucosal glands in *RUNX3^R122C/R122C^* mice were dominated by spasmolytic polypeptide/wound healing factor TFF2-expressing cells, unlike wildtype mice. The dramatic increase in rapidly proliferating isthmus stem/progenitor cells in the corpus of the *RUNX3^R122C/R122C^* mice was accompanied by mucous neck cell hyperplasia and massive reductions of pit, parietal and chief cell populations [41]. Whereas expression of stem cell factor Sox9 was mainly located in the isthmus of wildtype mice, the metaplastic glands of *RUNX3^R122C/R122C^* mice were characterized by elevated Sox9 expression. It was suggested that while metaplasia is likely reversible in normal tissues, chronic inflammation may promote the establishment of metaplasia into a stable and potentially precancerous lesion [72]. Interestingly, prominent inflammatory infiltrates were observed in *RUNX3^R122C/R122C^* mice on the C57BL/6 background [41] and this contrasted with negligible inflammatory cell infiltration in 6-month-old *Runx3*-deficient mice on the BALB/c background [42]. M2 macrophages, an immune cell type that promotes preneoplastic metaplasia [73,74], were increased in the gland base of *RUNX3^R122C/R122C^* mice [41].

The transcriptomic profile of the *RUNX3^R122C/R122C^* corpus tissue showed enrichment of the early gastric cancer gene signature, as well as pathways related to the cell cycle and the inflammatory response [41]. The top ranked upregulated pathway was the interferon-γ response. We note that interferon-γ, a cytokine associated with autoimmunity and infection, might promote the transition to metaplasia and early-stage cancer [75]. Other upregulated pathways include MYC and KRAS, which are reminiscent of observations from *Runx3* KO mouse models in the intestine and lung [51,60].

Organoids derived from the RUNX3^R122C/R122C^ mice formed with significantly higher efficiencies than organoids from wild-type mice [41]. While this observation indicates that the enhanced stem cell activity in *RUNX3^R122C/R122C^* cells was independent of immune cells, inflammation-induced stem cell proliferation in *RUNX3^R122C/R122C^* remains a distinct possibility. Chief cells have been shown to reprogram into SPEM [72]. Taken together with the dramatic increase of isthmus stem cells in *RUNX3^R122C/R122C^* mice, it is likely that hyperproliferation of isthmus stem cells, together with reprogramming of chief cells, resulted in the emergence of SPEM, as characterized by an antral type mucosa, albeit without mature foveolar cells [41].

How the tissue stem cell balances between quiescence, proliferation and differentiation is frequently hampered by the lack of definitive stem cell markers. Given that we had recently identified cytoskeletal scaffold protein *IQGAP3* as a specific marker for proliferating isthmus stem cells in the corpus [76], and found that IQGAP3 is dramatically induced in the isthmus region of *RUNX3^R122C/R122C^* gastric pit, we used *IQGAP3* as a molecular tool to isolate and characterize *RUNX3^R122C/R122C^* stem cells. Transcriptomic profiling has indicated that enrichment of the cell-cycle-related gene signatures, such as MYC and E2F targets, in the isthmus stem cells of *RUNX3^R122C/R122C^* mice promote the proliferation of precancerous lesions [41] (Figure 2).

The highly proliferative stem cells, their expansion and impaired ability to terminally differentiate may be attributed to the inability of the RUNX3^R122C^ mutant protein to bind BRD2 and regulate the restriction point [41]. The proliferative cells detected at the gland base of *RUNX3^R122C/R122C^* mice potentially reflect the dedifferentiation of mature chief cells and re-entry into the cell cycle [41,77,78] (Figure 3). Curiously, despite exhibiting a precancerous phenotype, *RUNX3^R122C/R122C^* mice older than 1.5 years did not develop gastric cancer [41]. While additional genomic alterations are likely necessary for malignant transformation, an alternative reason may reside in the function of RUNX3 in various immune cell types [79] and the altered ability of *RUNX3^R122C/R122C^* mice to shape the local immune microenvironment. Further studies of *RUNX3^R122C/R122C^* mice will provide insights on the function of RUNX3 on the intrinsic self-renewal capacity of stem cells, as well as their communication with the immune environment during cancer initiation.

## 7. The Potential Interplay between *RUNX1* and *RUNX3*

As discussed earlier, *RUNX1* is a stem cell factor that is highly expressed in several epithelial tumors [13]. *Runx1* KO mice indicate the importance of *Runx1* for tumor formation in hair follicle stem cells [13]. *Runx1* promotes mouse skin squamous tumor formation, potentially through repression of *Cdkn1a* and promotion of *Stat3* activation. Conversely, *RUNX1* may exert tumor suppressor activity in the mouse intestine and human luminal breast cancer [9,10,13,80,81].

*Runx1* expression in hematopoietic stem cells is, in part, regulated by an enhancer, termed the +24 conserved noncoding element (hereafter, referred to as *eR1*) [82,83]. More recently, we found that *eR1* drives the expression of Runx1 in stem cells at the isthmus of the stomach corpus, as well as a small population of terminally differentiated chief cells at the gland base [84]. When *eR1* was used to target oncogenic *Kras^G12D^* expression to the stem cells, pseudopyloric metaplasia were induced [84]. It is interesting that the majority of *Runx1*-positive cells colocalized with proliferation marker Ki67 expression at the isthmus [84]. Is Runx1 serving a pro-proliferative role in isthmus stem cells? As yet, the function of *Runx1* in proliferating isthmus stem cells is unclear. In one scenario, the aberrant expression of *Runx1* promotes tumorigenesis in stem cells and is restrained by *Runx3*. In the second scenario, *Runx3* and *Runx1* serve complementary roles to ensure stem cell homeostasis. The elevated stem cell population in the *RUNX3^R122C/R122C^* mouse model may reflect an imbalance of Runx1 and Runx3 activity [41]. *eR1* affords the opportunity of manipulating *Runx3* expression in *Runx1*-positive stem cells and is a potentially useful tool to study the relationship between *Runx1* and *Runx3* in epithelial stem cells.

## 8. Discussion

Despite more than two decades of RUNX3 research, the function of RUNX3 in epithelial stem cells remains unclear. This may be due to the low expression of *RUNX3* in epithelial cells, relative to immune cells [34,35]. While there are indications that *RUNX3* expression is subjected to environmental cues, be they developmental or stress-related [39,85], how *RUNX3* gene expression is regulated during epithelial tissue homeostasis, tissue damage and regeneration remain hazy. Moreover, our knowledge of RUNX3’s influences on the Ras, TGFβ and Wnt signaling pathways in stem cells during these periods is incomplete. Although RUNX proteins have been linked to epigenetic memory and asymmetric division, the roles of RUNX3 in these fields remain to be elucidated. We will also need to consider how RUNX3 modulates immune cell function to regulate stem cell behavior during normal growth, inflammatory response and damage-induced regeneration.

RUNX3 serves context-dependent roles in cancer [8]. The frequent epigenetic inactivation of *RUNX3* in cancer [8] and the paradoxical elevated *RUNX3* expression in metastatic pancreatic cancer [36] indicate that tight control of RUNX3’s activities are crucial for normal growth. As discussed, it is tumor-suppressive during early-stage cancer and oncogenic in metastatic cancer. It is of interest to understand what promotes epigenetic silencing of RUNX3 early during tumorigenesis and how *RUNX3* is abnormally upregulated in certain metastatic cells. RUNX3 strongly influences core cellular processes (e.g., cell cycle and apoptosis) and signaling pathways (e.g., TGFβ, Wnt and Ras) that are frequently altered in cancer [17]. The fact that these pathways are critically involved in stem cell proliferation and differentiation prompts the question: is RUNX3 a stem cell gatekeeper?

The *Runx3* heterozygous KO and *RUNX3^R122C/R122C^* mutant mouse models have indicated functional roles for RUNX3 in gastric epithelial homeostasis and stem cell regulation [41,42]. *Runx3* deficiency resulted in hyperproliferation, stem cell expansion and dedifferentiation in the stomach corpus [41,42]. It is important to ascertain to what extent the molecular model drawn in the stomach can be used in other tissue types. Notably, *Runx3*-deficient mouse models are cancer prone in different tissue types, such as the intestine, mammary gland, stomach and lung [42,51,58,60]. It is, thus, reasonable to consider commonalities in RUNX3’s functions across diverse tissue stem cells. Moreover, unlike many tumor suppressor genes, *RUNX3* is mainly inactivated by epigenetic aberrations and not genetic mutations. Unlike mutations, epigenetic alterations are reversible. Whether re-activation of silenced *RUNX3* using demethylating agents (e.g., azacytidine and decitabine) leads to terminal differentiation and tumor regression remain to be determined. Conversely, the aberrant upregulation of RUNX3 in metastatic cancer may be targeted therapeutically by small molecule inhibitors, such as the pyrrole-imidazole polyamide Chb-M’, benzodiazepine Ro5-3335, as well as 2-pyridyl benzimidazole AI-4-57 and derivatives [86,87,88,89]. Chb-M’ targets the RUNX consensus DNA-binding sequence, while AI-4-57 and Ro5-3335 target the RUNX–CBFβ interaction [86,87,88,89].

While early work revealed that RUNX3 inhibits cancer initiation by inhibiting cell cycle entry and inducing apoptosis [47,48], it is tempting to hypothesize that one of RUNX3’s tumor suppressor properties may reside in its function as a reprogramming barrier or through maintenance of a stable differentiated state and that *Runx3* inactivation correlates with increased plasticity and/or the acquisition of a stem-cell-like state. The abilities of RUNX3 to restrict cell cycle progression through induction of the cyclin-dependent kinase inhibitor *CDKN1A* and to induce apoptosis through induction of *BIM* may limit reprogramming [47,48]. The p53 pathway has been shown to be a determinant of reprogramming efficiency [90,91]. RUNX3 was reported to enhance p53-mediated transcriptional activation [85]. Furthermore, RUNX3 has been reported to transcriptionally activate *ARF* [60]. Given that ARF is an antagonist of MDM2, the E3-ubiquitin ligase responsible for p53 degradation, *ARF* can function as a barrier to cell reprogramming [90]. *ARF* has been implicated in the regulation of stem cell population—enforced expression of *ARF* in hematopoietic stem cells led to p53-dependent cell death [37,92]. Clearly, further studies on the RUNX3–p53 cooperation would help illuminate how cells protect against lineage plasticity and cancer development.

The complex crosstalk between RUNX3 and multiple signaling pathways is likely to strongly influence epithelial and immune phenotypes. Going forward, mechanistic studies on how RUNX3 directs stem cell behavior in response to the cues from the immune microenvironment are likely to yield insights on the communication of epithelial cells with immune cells in the defense against tumorigenesis.

## Figures and Tables

**Figure 1 cells-12-00408-f001:**
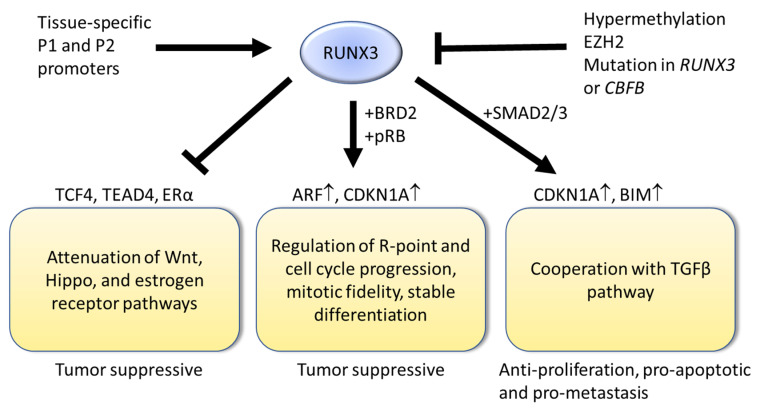
Potential mechanisms for RUNX3 as a gatekeeper in stem cells. The *RUNX3* gene is regulated by the tissue-specific P1 and P2 promoters. In cancer cells, RUNX3 may be inactivated by hypermethylation, EZH2-mediated chromatin repression or somatic mutations in RUNX3 or, obligate partner, CBFB. RUNX3 inhibits the activities of TCF4, TEAD4 and ERα proteins to attenuate the oncogenic Wnt, Hippo and estrogen receptor pathways, respectively. Early in the cell cycle, RUNX3 interacts with BRD2 and pRB proteins to induce *ARF* and *CDKN1A* gene expression and thereby regulate R point. RUNX3 interacts with TGFβ effectors to cooperate with the dualistic TGFβ pathway.

**Figure 2 cells-12-00408-f002:**
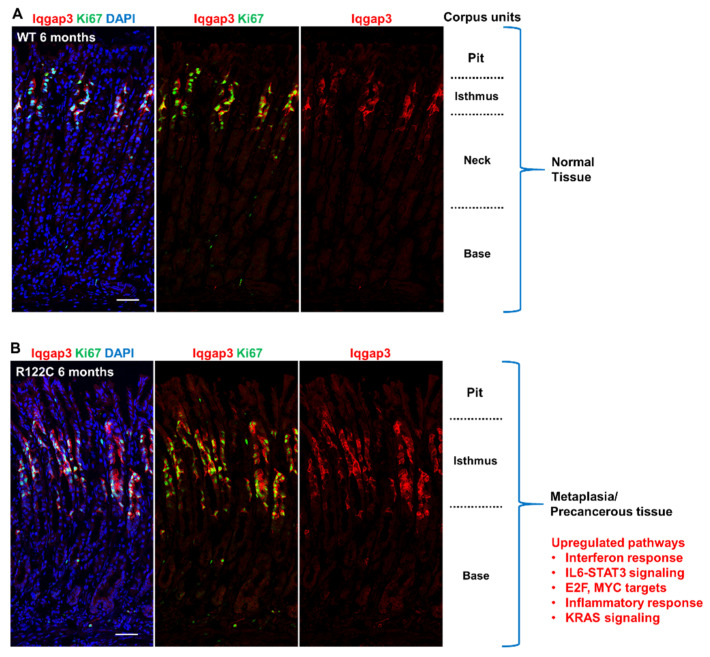
*RUNX3^R122C/R122C^* mice exhibit increased isthmus stem cell proliferation. Immunofluorescence staining of corpus units in wild-type and *RUNX3^R122C/R122C^* mice. Proliferating isthmus stem cells are marked by stem cell marker IQGAP3 (red) and proliferation marker Ki67 (green). DNA is stained by DAPI. (**A**) Wild-type mice at 6 months of age. (**B**) *RUNX3^R122C/R122C^* mice at 6 months of age show precancerous tissue with the indicated upregulated pathways [41].

**Figure 3 cells-12-00408-f003:**
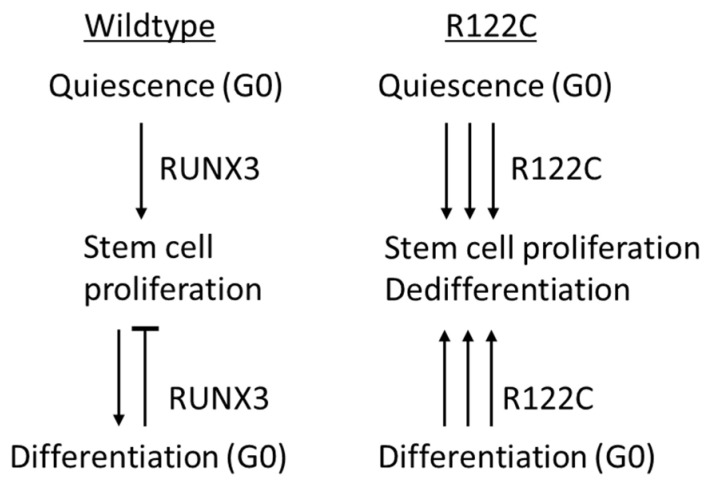
Proposed model for stem cell dysregulation in *RUNX3^R122C/R122C^* mouse stomach. Left, wild-type RUNX3 regulates restriction point, maintains stem cell homeostasis and promotes stable differentiation. Right, in *RUNX3^R122C/R122C^* mice, R point regulation is dysfunctional, resulting in activation and proliferation of stem cells.

## Data Availability

Not applicable.

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
