# Peer review of "RUNX3 in Stem Cell and Cancer Biology"

_cells, 2023, doi:10.3390/cells12030408_

Round 1
Reviewer 1 Report
Major points
1. Line 157, 224, 242: as the insertion of the reference 41 and 42 implies, the potential issues or insights of using different mouse strains in studying the roles of Runx proteins in carcinogenesis can be further discussed since this is an important point. It can be described in the early part of this section 4.
2. Line 172: it might be useful to add the reference “Hor et. al., Cell rep. 2014, 10.1016/j.celrep.2014.06.003” and discuss a little more about the mechanistic view of H. pylori and Runx3. Especially, why IL-23A is expressed from epithelial cells under the influences of Runx3 and the potential effects on innate immunity within the stomach could be discussed.
3. Line 215: are there studies investigated any new binding resulted from the mutation not just the loss of the canonical binding? Also, is it possible that R122C mutation affects the amounts or stability of Runx3 in addition to the DNA binding specificity (binding ability to CBFb?)?
Minor points
1. Line 148: it might be good to add a few words to describe “Apc”.
2. Line 154: it might be good to define “HMG” within parenthesis.
3. Line 159: “the parietal cell” and “chief cell” might need the definition for the readers who are not familiar with gastrointestinal epithelial cells.
4 .Line 192: a brief explanation of BRD2 and pRB might help the readers to understand this section.
5. Line 221: “significant consequences” might require a few examples such as binding to CBFb etc.
6. Line 247: any gene related to H. pylori?
7. Line 258: the definition of SPEM” might be necessary.
8. The discussion section could include a brief description about potential pharmacological intervention of Runx or Runx-mediated pathways.
Author Response
Responses to Reviewer 1
Major points
Comment 1: Line 157, 224, 242: as the insertion of the reference 41 and 42 implies, the potential issues or insights of using different mouse strains in studying the roles of Runx proteins in carcinogenesis can be further discussed since this is an important point. It can be described in the early part of this section 4.
Response: We have included a discussion of the use of different mouse strains at the beginning of section 4 (lines 130-133):
‘Mouse knockout models have been instrumental in understanding the role of RUNX3 during carcinogenesis. While it should be noted that Runx3 deficiency in different mouse strains (eg. BALB/c and C57BL/6) demonstrated phenotypic variability with respect to the inflammatory response, all revealed susceptibility to preneoplastic changes.’
Comment 2: Line 172: it might be useful to add the reference “Hor et. al., Cell rep. 2014, 10.1016/j.celrep.2014.06.003” and discuss a little more about the mechanistic view of H. pylori and Runx3. Especially, why IL-23A is expressed from epithelial cells under the influences of Runx3 and the potential effects on innate immunity within the stomach could be discussed.
Response: Thank you for the helpful suggestion. We have cited Hor et al (2014) and discussed RUNX3 and IL23A in the stomach. Please see lines 186-190:
‘Notably, RUNX3 directly activates the transcription of one of the key regulators of innate immunity, IL23A [55]. This ability to upregulate IL23A expression is strongly enhanced by TNF-α/NF-kB stimulation and H. pylori infection, thus indicating the involvement of RUNX3 during infection of gastric epithelial cells and its potential protective roles in the inflammatory response and pathogen clearance [55].’
- Line 215: are there studies investigated any new binding resulted from the mutation not just the loss of the canonical binding? Also, is it possible that R122C mutation affects the amounts or stability of Runx3 in addition to the DNA binding specificity (binding ability to CBFb?)?
Response: The stability of the R122C mutant protein, its DNA binding and CBFβ binding abilities are currently unknown. Nevertheless, the Reviewer has made very valid suggestions for future work. These points are highlighted in lines 251-256:
‘This impaired ability to inhibit TCF4 and TEAD4 indicates that the RUNX3R122C mutation – independent of its altered affinity for DNA – might have significant consequences on Wnt and TEAD signaling activities. It is currently unknown whether the RUNX3R122C mutant binds sequences other than the consensus RUNX sequence. Moreover, it remains to be seen whether the R122C mutation affects RUNX3 protein stability and binding to CBFβ.’
Minor points
- Line 148: it might be good to add a few words to describe “Apc”.
Response: Adenomatous polyposis coli (Apc) is described in lines 152-153 and 160-168:
‘APC is a key negative regulator of the Wnt pathway. It provides a scaffold for the β-catenin destruction complex and is important for rapid β-catenin turnover [52]. The high frequency of APC gene mutations in colorectal cancers suggests that APC dysfunction and subsequent elevated Wnt activity are early and/or initiating events in colorectal cancer [52]. Runx3 deficient mice were thus compared with the ApcMin/+ mouse model, which harbors a dominant loss of function mutation at the Apc gene. At 65 weeks of age, the frequency of adenoma development in the small intestine of Runx3+/- mice was comparable to that of ApcMin/+ mice with the same BALB/c back-ground [51].’
- Line 154: it might be good to define “HMG” within parenthesis.
Response: HMG is described as ‘high mobility group’ within parenthesis.
- Line 159: “the parietal cell” and “chief cell” might need the definition for the readers who are not familiar with gastrointestinal epithelial cells.
Response: The parietal cell is now described as ‘acid secreting’, while chief cell is described as ‘digestive enzyme-secreting and terminally differentiated’. Please see lines 173-174.
4 .Line 192: a brief explanation of BRD2 and pRB might help the readers to understand this section.
Response: Explanations for BRD2 and pRB are provided in lines 208-210 and 217-223, respectively.
‘Bromodomain proteins are considered to be chromatin ‘readers’, which recruit enzymes that modify chromatin to regulate gene expression [61]. How RUNX3 influences chromatin modelling during the various phases of the cell cycle remains to be determined.’
‘The retinoblastoma susceptibility gene (RB) is frequently mutated in a wide range of cancer types [64]. While pRB is best known – through its inhibition of E2F transcription factors – for its role in regulating G1-S transition, it is now considered to be a transcription co-factor that binds and modulates the activities of numerous transcription factors as well as an adaptor protein that recruits chromatin modelling enzymes to target genes [64]. pRB has been reported to regulate differentiation, maintain permanent cell cycle arrest as well as chromosomal stability [64].’
- Line 221: “significant consequences” might require a few examples such as binding to CBFb etc.
Response: We have explained the significant consequences in more detail (please see lines 251-256)
‘This impaired ability to inhibit TCF4 and TEAD4 indicates that the RUNX3R122C mutation – independent of its altered affinity for DNA – might have significant consequences on Wnt and TEAD signaling activities. It is currently unknown whether the RUNX3R122C mutant binds sequences other than the consensus RUNX sequence. Moreover, it remains to be seen whether the R122C mutation affects RUNX3 protein stability and binding to CBFβ.’
- Line 247: any gene related to H. pylori?
Response: Aside from an early gastric cancer signature, we do not observe any specific gene related to H.pylori.
- Line 258: the definition of SPEM” might be necessary.
Response: SPEM was earlier described in lines 264 to 268: ‘a precancerous phenotype known as spasmolytic polypeptide-expressing metaplasia (SPEM) in the stomach corpus [41]. The elongated fundic metaplastic mucosal glands in RUNX3R122C/R122C mice were dominated by spasmolytic polypeptide / wound healing factor TFF2-expressing cells, unlike wildtype mice.’
- The discussion section could include a brief description about potential pharmacological intervention of Runx or Runx-mediated pathways.
Response: We have provided a discussion of potential pharmacological intervention in lines 383-390.
Whether re-activation of silenced RUNX3 using demethylating agents (eg. azacytidine and decitabine) leads to terminal differentiation and tumor regression remain to be determined. Conversely, the aberrant upregulation of RUNX3 in metastatic cancer may be targeted therapeutically by small molecule inhibitors, such as the pyrrole-imidazole polyamide Chb-M', benzodiazepine Ro5-3335, as well as 2-pyridyl benzimidazole AI-4-57 and derivatives [86-89]. Chb-M’ targets the RUNX consensus DNA‐binding sequence, while AI-4-57 and Ro5-3335 target the RUNX-CBFβ interaction [86-89].

Reviewer 2 Report
This is a very well written comprehensive review from the authors and elegantly encapsulates the current working knowledge of RUNX3 in stem-cell biology and how this links with cancer. Particular emphasis is given to the R122C mutant mouse model which gives a fresh outlook on this topic. It was a pleasure to read this review.
Minor comments:
Abstract line 10 'plays' should be 'play'
lines 42-43: check English of the sentence - homozygous Runx1 KO mice were unable to generate of hematopoietic stem cells and showed embryonic lethality
Figure 2 has been adapted from previous work from the authors. The journal can check copyright but perhaps the authors have additional representative examples of these results that could be shown?
Line 354: additional 'that'
Author Response
Responses to Reviewer 2
Minor comments:
Abstract line 10 'plays' should be 'play'
Response: Thank you, we have made the correction.
lines 42-43: check English of the sentence - homozygous Runx1 KO mice were unable to generate of hematopoietic stem cells and showed embryonic lethality
Response: Thank you, we have corrected the English.
Figure 2 has been adapted from previous work from the authors. The journal can check copyright but perhaps the authors have additional representative examples of these results that could be shown?
Response: We have replaced the figure. As this figure is not copyrighted, we have removed the note on licensing agreement.
Line 354: additional 'that'
Response: Thank you, we have removed the additional ‘that’.
Round 2
Reviewer 1 Report
Thank you for addressing and incorporating the raised comments.